# Trabecular Bone Assessment Using Magnetic-Resonance Imaging: A Pilot Study

**DOI:** 10.3390/ijerph17249282

**Published:** 2020-12-11

**Authors:** Lauren Bohner, Pedro Tortamano, Norbert Meier, Felix Gremse, Johannes Kleinheinz, Marcel Hanisch

**Affiliations:** 1Department of Cranio-Maxillofacial Surgery, University Hospital Muenster, 48149 Muenster, Germany; johannes.kleinheinz@ukmuenster.de (J.K.); marcel.hanisch@ukmuenster.de (M.H.); 2Department of Prosthodontics, School of Dentistry, University of São Paulo, São Paulo 0508-000, Brazil; tortamano@usp.br; 3Institute of Clinical Radiology, University Clinics Muenster, 48149 Muenster, Germany; norbert.meier@ukmuenster.de; 4Department of Experimental Molecular Imaging, Helmholtz Institute, RWTH Aachen University, 52074 Aachen, Germany; fgremse@ukaachen.de

**Keywords:** dental implants, magnetic-resonance imaging, microCT

## Abstract

The aim of this study was to assess trabecular bone morphology via magnetic-resonance imaging (MRI) using microcomputed tomography (µCT) as the control group. Porcine bone samples were scanned with T1-weighted turbo spin echo sequence imaging, using TR 25 ms, TE 3.5 ms, FOV 100 × 100 × 90, voxel size 0.22 × 0.22 × 0.50 mm, and scan time of 11:18. µCT was used as the control group with 80 kV, 125 mA, and a voxel size of 16 µm. The trabecular bone was segmented on the basis of a reference threshold value and morphological parameters. Bone volume (BV), Bone-volume fraction (BvTv), Bone specific surface (BsBv), trabecular thickness (TbTh), and trabecular separation (TbSp) were evaluated. Paired *t*-test and Pearson correlation test were performed at *p* = 0.05. MRI overestimated BV, BvTv, TbTh, and TbSp values. BsBv was the only parameter that was underestimated by MRI. High statistical correlation (r = 0.826; *p* < 0.05) was found for BV measurements. Within the limitations of this study, MRI overestimated trabecular bone parameters, but with a statistically significant fixed linear offset.

## 1. Introduction

As far as dental-implant planning is concerned, special attention is given to trabecular-bone assessment. Although adequate bone volume is a requisite for peri-implant health, it has been proven that bone quality may also play an important role in dental-implant outcomes. According to the classification of Lekholm and Zarb [1], the jaw bone may either present cortical bone with different thickness or trabecular bone with variable microarchitecture. In this sense, recent studies showed that cortical thickness influences the primary stability of dental implants [2,3]. However, successful implant treatment is related not only to bone volume, but also to the micromorphology of trabeculae, and how they are arranged and connected to each other [4].

Cone-beam computed tomography (CBCT) is the method of choice for bone assessment. This technique has the advantage of three-dimensional (3D) reconstruction with sufficient spatial resolution compared to conventional methods [5,6,7]. However, CBCT indication is limited due to its ionizing nature [8]. Thus, in order to offer a safe method for bone assessment, the use of alternative techniques, such as magnetic-resonance imaging (MRI), was extensively explored [9,10,11,12,13].

The use of MRI for implant planning and monitoring was also investigated [11,12,14,15,16,17,18,19,20,21,22]. It shows potential to evaluate bone dimension and visualize bone defects prior to the implant placement [11,22]. Although artifacts may occur due to the presence of oral tissue and dental materials, they seem to be localized and, in several cases, negligible for diagnosis [23].

Moreover, the medical literature showed its potential to determine bone quality of osteoporotic patients [24]. However, the assessment of bone quality for dental-implant planning has still not been reported. The aim of this study was to assess trabecular-bone morphology through MRI, using microcomputed tomography (µCT) as the control group.

## 2. Materials and Methods

Porcine ribs acquired from a local butchery for a previous study [25] were used in this study. Samples were scanned with MRI, and µCT scans were used as reference values. The study was approved by the Ethics Committee of Hospital University Münster (protocol number 2017-629-f-N).

A pilot study was performed, and sample size was calculated using GPower 3.1 (University of Düsseldorf) [26]. Considering a two-tailed *t*-test, with a power of 80% and significance level at 0.05, at least 7 samples were required to detect a mean difference of 0.26 ± 0.19 mm^3^ (mean ± standard deviation) in trabecular-bone volume measured by µCT and MRI. In order to control intrarater reliability, measurements were performed twice by the same examiner.

Briefly, porcine ribs were prepared by removing soft tissue and periosteum, and then sectioned into bone pieces measuring approximately 3 cm in width. On each sample, a conical perforation was prepared according to the manufacturer instructions using dental-implant surgical drills (Straumann). For this study, dental implants were not placed. Instead, the perforation was used as a reference marker for the determination of the measurement site. All samples were immersed into a 3.5% formaldehyde bath 4 weeks prior to the study.

As a gold-standard control, each bone sample was scanned with µCT (SkyScan 1272; Bruker, Kontich, Belgium) with 80 kV, 125 mA, and a voxel size of 16 µm. For MRI scanning, each sample was inserted into a plastic conical container filled with ultrasound gel. T1-weighted turbo spin echo sequence imaging was performed using a whole-body 3T magnetic-resonance system (Philips, Healthcare System) with 8-channel SENSE-foot/Ankle coil, TR 25 ms, TE 3.5 ms, FOV 100 × 100 × 90, voxel size 0.22 × 0.22 × 0.50 mm, and a scan time of 11:18. Tissue with short T_2_ relaxation times was presented as bright values.

Trabecular-bone segmentation and assessment were performed using imaging software (Imalytics Preclinical, Gremse-IT GmbH) [27]. First, images were reoriented with the center of perforation on a three-coordinate axis (x,y,z) as reference. A center marker was set at the apex region exactly 10 mm from the middle-diameter of the perforation. This marker was used as reference to define the center of the volume of interest (VOI). Using an automatic tool, a conical VOI measuring 5 × 10 mm (diameter × height) was defined (Figure 1).

In order to define the gray-scale value that separated the trabecular bone from the medullary space, a mean threshold value was calculated for each image. Trabecular-bone segmentation used this threshold value as reference, and distance mapping was employed to refine the segmentation on the basis of morphological features (Figure 2). Table 1 indicates the bone parameters calculated in this study.

Statistical analysis was performed using SPSS 0.26 software (IBM, Armonk, NY, USA) and reviewed by an independent statistician. Data normality was evaluated by Shapiro–Wilk test, and intrareader reliability was assessed using Cronbach’s test. A two-tailed paired *t*-test was used to compare the mean values measured by µCT and MRI, whereas the relation between groups was assessed by linear regression and Pearson’s correlation test. The statistical-significance level was considered at *p* ≤ 0.05.

## 3. Results

Data presented adherence to Gaussian distribution, and were therefore described by mean and standard deviation. Intraclass reader reliability was high (0.96; *p* < 0.05). Table 2 and Table 3 show the statistical analysis.

In general, MRI overestimated trabecular-bone parameters in comparison to µCT. Bone specific surface (BsBv) was the only parameter that was underestimated by MRI. This underestimation, in combination with the overestimation of further parameters, is usually associated with a denser bone [4]. All differences were statistically significant at *p* = 0.05. High statistical correlation (r = 0.826; *p* < 0.05) was found for BV measurements indicating a variance of r^2^ = 0.683 (*p* = 0.02) using linear-regression model y = 1.04x + 0.30.

## 4. Discussion

MRI was proposed for dental-implant planning since it depicts accurately anatomical structures relevant for the surgical procedure. Previous studies focused on digital planning to determine the implant position, showing that the technique is capable of determining BV within clinical requirements [28,29]. However, to the authors’ knowledge, this is the first study using MRI to assess trabecular-bone morphology.

The findings of this study showed that MRI overestimated bone parameters when compared to the gold-standard µCT. Thus, it can indicate that the bone is denser than it really is. Nonetheless, especially bone-volume determination was highly correlated with the reference values. Through mathematical formulation, it was possible to predict that, for each increase in CBCT-determined BV, an increase of 1.04 mm^3^ occurred on the basis of T1-weighted MRI. This linear relation explains 68.3% of the variation between groups.

Porcine ribs were chosen due to their similarity with the human maxillary bone [26]. However, these results must be interpreted with caution since bone anatomical variation and interference of the surrounding tissue on T_2_ relaxation times were not considered. In addition, results may vary according to the scan protocol and the clinical situation. For instance, spatial resolution can affect the visualization of anatomical structures [15,21].

Furthermore, artifacts provided from structures, as metallic restorations, dental implants, or the simple interface between air and mucosa [19], were not considered. Clinically, it is expected that other factors, such as patient movements, could affect the image, resulting in lower image quality. Thus, although these preliminary results can support the use of MRI to the understanding of bone architecture prior to implant placement, they cannot be applied to the clinical field. Further studies are required before drawing a conclusion regarding bone-quality assessment using MRI.

Although this analysis focused on dental-implant planning, where a three-dimensional bone evaluation is required [11,12,22], assessment of the bone microarchitecture is well-known in the medical literature. For instance, MRI was shown to be a technique for monitoring systemic bone diseases since it allows for recognizing changes on trabecular-bone morphology [21]. Thus, developing specific protocols for the monitoring of bone-remodeling processes could benefit different medical fields [30,31].

The main limitations of this methodology are the small sample size and the absence of inter-reliability analysis to validate the measurements. Future studies should include MRI assessment considering clinical conditions instead of porcine bones. Moreover, further analysis should be conducted to determine whether the bone-quality assessment corresponds with current bone classification. In summary, this initial study shows that the use of MRI for the assessment of the trabecular bone is possible, provided that an accurate and reliable MRI protocol is used. Future studies should look for ways to increase MRI spatial resolution while using a clinical setup and a higher sample size.

## 5. Conclusions

Within the limitations of this study, MRI overestimated trabecular-bone parameters, but with a statistically significant fixed linear offset. Further studies are required to determine the clinical feasibility of MRI for trabecular-bone assessment.

## Figures and Tables

**Figure 1 ijerph-17-09282-f001:**
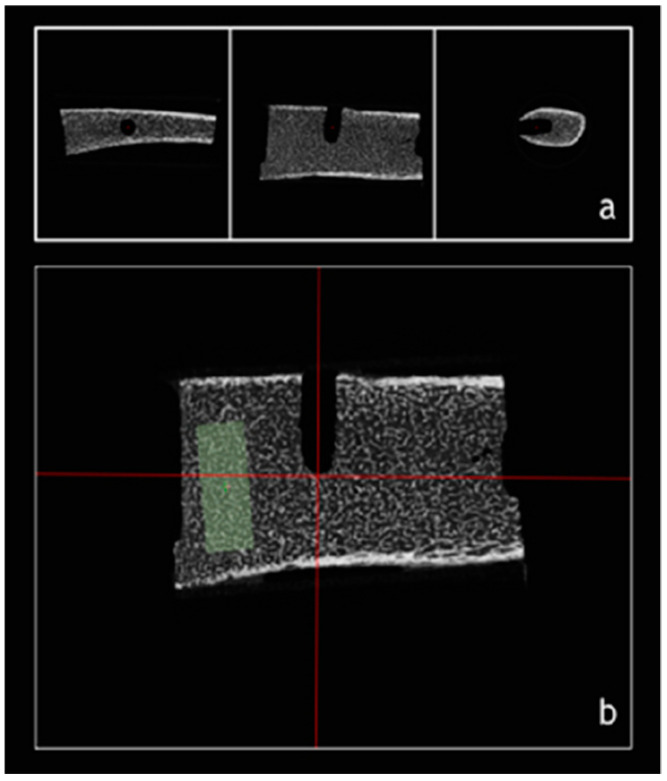
Determination of volume of interest (VOI). (**a**) From right to left: Axial, coronal, and sagittal cuts of microcomputed tomography (µCT) images. (**b**) Using reference lines, perforation midcenter was determined. and VOI was segmented.

**Figure 2 ijerph-17-09282-f002:**
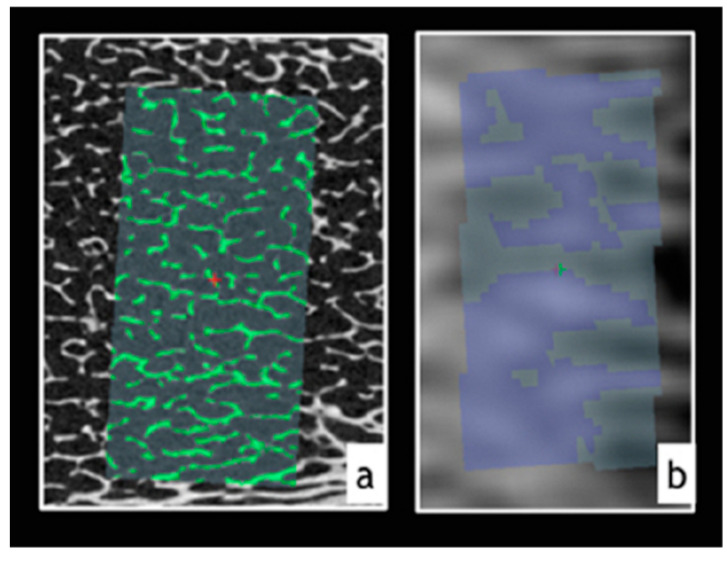
Segmentation of trabecular bone. Green and blue regions represent trabecular bone with (**a**) µCT segmentation and (**b**) magnetic-resonance-imaging (MRI) segmentation, respectively.

**Table 1 ijerph-17-09282-t001:** Bone measurement parameters.

Measurement	Abbreviation	Measurement Unit	Description
Trabecular bone volume	BV	mm^3^	Trabecular bone volume at VOI, determined by gray values above threshold value
Total bone volume	TV	mm^3^	Total bone volume (VOI)
Trabecular volume fraction	BvTv	%	Ratio between trabecular and total bone volume
Bone specific surface	BsBv	1/cm	Bone surface to bone volume ratio
Trabecular thickness	TbTh	cm	Trabecular bone thickness determined by distance between bone surface above threshold value
Trabecular separation	TbSp	cm	Mean distance between trabeculae, determined by gray values under threshold value

**Table 2 ijerph-17-09282-t002:** Descriptive data of bone measurements acquired by microcomputed tomography (µCT) and magnetic-resonance imaging (MRI).

Bone Measurements	Imaging Device	Mean ± SD
**BV (mm^3^)**		
	µCT	0.51 ± 0.22
	MRI	0.82 ± 0.28 *
**BvTv (%)**		
	µCT	29.48 ± 7.95
	MRI	48.86 ± 11.66 *
**BsBv (1/cm)**		
	µCT	146.23 ± 42.95
	MRI	30.06 ± 8.50 *
**TbTh (cm)**		
	µCT	0.25 ± 0.07
	MRI	1.28 ± 0.89 *
**TbSp (cm)**		
	µCT	0.58 ± 0.07
	MRI	1.19 ± 0.21 *

* statistically significant difference at *p* ≤ 0.05.

**Table 3 ijerph-17-09282-t003:** Statistical analysis.

Paired *T*-Test					Pearson Correlation	
Bone Measurements	*t*-Value	df	95% CI		r	*p* Value
			Inf	Sup		
**BV**	−5.8	6	−0.46	−0.189	0.826	0.01
**BvTv**	−4.49	6	−29.92	−8.81	0.274	0.27
**BsBv**	7.93	7	81.53	150.80	0.083	0.43
**TbTh**	−3.17	6	−1.82	−0.23	0.522	0.11
**TbSp**	−8.45	6	−0.79	−0.43	0.464	0.14

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
