# Peer review of "Trabecular Bone Assessment Using Magnetic-Resonance Imaging: A Pilot Study"

_ijerph, 2020, doi:10.3390/ijerph17249282_

Round 1

Reviewer 1 Report

This is a quite interesting manuscript on the assesment of trabecular bone pattern using MRI. This topic is certainly of interest for the dental implantologist who want to reduce the invasivness of the CT exam.

The introduction is simple, appropriate, and well written. However, it could be improved by referring the different classifications of bone quality useful for implant positioning (Norton and Gamble classification, Lekholm and Zarb…) and already present in literature. 

The abstract is synthetic and explicative. However, I suggest the author to add in the abstract that is synthetic and explicative, in the aim of the study that “a micro computed tomography (μ CT) was used as control group”.

Main criticism: Page 2 Line 54-55: it is definded that in your study in order to control intra-rater reliability, measurements were performed twice by the same examiner. For this work, it is more important the inter-rater reliability. A method error should be checked and repeated measurement from different operators, in different moments, would avoid this bias. It would be adequate to state another article that uses the same methodology to assert a similar statement. 

Data and analyses are appropriately presented. 

Discussion could be improved. As a dental implantologist, I would know if the use of MRI instead of CBCT could help me in the classification of bone quality. May the MRI values you reported, be useful to understand the quality of bone, although there is a significant fixed linear offset and your study is on animals and not people? 

And how could your results be compared with Lekholm, Zarb or Norton and Gamble classification or, if not, with other classifications on animals? Please discuss

Please, check the reference format in the text as author guidelines indicated (e.g. Line 69, 117… References have to be indicated always before the full-stop). 

Please correct the Author Contributions statement by eliminating the final phrases.

Author Response

Dear reviewers,

We would like to thank you for reviewing our manuscript and for helping us to improve it for possible publication. We agreed with all suggestions and we tried to improve the manuscript accordingly. All changes were highlighted throughout the manuscript and a point-by-point response is given below:

Reviewer 1:

This is a quite interesting manuscript on the assesment of trabecular bone pattern using MRI. This topic is certainly of interest for the dental implantologist who want to reduce the invasivness of the CT exam.The introduction is simple, appropriate, and well written. However, it could be improved by referring the different classifications of bone quality useful for implant positioning (Norton and Gamble classification, Lekholm and Zarb…) and already present in literature.

Response:

We added a sentence citing the Classification of Lekholm and Zarb (lines 31-35):

“Although an adequate bone volume is a requisite for periimplant health, it has been proved that bone quality may also play an important role on dental implant outcomes. According to the Classification of Lekholm and Zarb [1], jaw bone may either present cortical bone with different thickness or trabecular bone with variable microarchitecture. In this sense, recent studies have already shown that cortical thickness influences the primary stability of dental implants [2,3].”

The abstract is synthetic and explicative. However, I suggest the author to add in the abstract that is synthetic and explicative, in the aim of the study that “a micro computed tomography (μ CT) was used as control group”.

Response:

We added this information on line 14.

Main criticism: Page 2 Line 54-55: it is definded that in your study in order to control intra-rater reliability, measurements were performed twice by the same examiner. For this work, it is more important the inter-rater reliability. A method error should be checked and repeated measurement from different operators, in different moments, would avoid this bias. It would be adequate to state another article that uses the same methodology to assert a similar statement. 

Thank you for your observation.

We agree that this is a limitation of this study, since inter-rater reliability is important to validate measurements. Considering the limitations of this study, we agreed to define the study as a pilot study and we changed the title accordingly (lines 2,3). In addition, we cited the methodological limitations (lines 145,146):

“The main limitations of this methodology are the small sample size, such as the absence of inter-reliability analysis to validate the measurements.”

Data and analyses are appropriately presented. Discussion could be improved. As a dental implantologist, I would know if the use of MRI instead of CBCT could help me in the classification of bone quality. May the MRI values you reported, be useful to understand the quality of bone, although there is a significant fixed linear offset and your study is on animals and not people?  And how could your results be compared with Lekholm, Zarb or Norton and Gamble classification or, if not, with other classifications on animals? Please discuss

Response:

Thank you for this important observation. We focused the Discussion on clinical relevance and cited the limitations of this study when considering a clinical situation (lines 128-139):

“Porcine ribs were chosen due to its similarity with human maxillary bone [26]. However, these results must be interpreted with caution, since bone anatomical variation and interference of surrounding tissues on T2 relaxation times were not taken in consideration. In addition, results may vary according to the scan protocol and to the clinical situation. For instance, spatial resolution can affect the visualization of anatomical structures [21,22].

Furthermore, it must be considered that artifacts provided from structures, as metallic restorations, dental implants, or the simple interface between air and mucosa [19], were not taken into account. Clinically, it is expected that other factors, as patient movements, could affect the image, resulting on a lower image quality. Thus, although these preliminary results can support the use of MRI to the understanding of bone architecture prior to the implant placement, they can not be applied to the clinical field. Further studies are required before drawing a conclusion regarding bone quality assessment using MRI. “

Furthermore, we do not believe that these preliminary results can be applied to a clinical situation. We addressed this issue on the conclusion as described below (lines 154,155):

“Further studies are required to determine the clinically feasibility of MRI for trabecular bone assessment.”

Please, check the reference format in the text as author guidelines indicated (e.g. Line 69, 117… References have to be indicated always before the full-stop). 

Response:

We corrected the references throughout the text.

Please correct the Author Contributions statement by eliminating the final phrases.

We deleted the final phrases.

Reviewer 2 Report

the title of your work is: "Assessment of trabecular bone for dental implant planning using magnetic resonance imaging" but in the paper the informations regarding the planning of dental implants are few. Could you consider to improve this part o modify the title?

The comparison between the two image acquisition methods is interesting, especially because MRI doesn't provide x-ray to patients.

Statistical analysis is carried out in a proper way.  

Please check again the typos and form, ex. line 117 missing bracket in bib.

I suggest to expand the field of this pilot study, introducing some evaluation of TMJ, where the MRI is widely used, you can use the following paper as reference:

Sambataro, S.; Cervino, G.; Bocchieri, S.; La Bruna, R.; Cicciù, M. TMJ Dysfunctions Systemic Implications and Postural Assessments: A Review of Recent Literature. J. Funct. Morphol. Kinesiol. 2019, 4, 58.

Please expand the conclusion, including some clinical implication of this work.

Author Response

Dear reviewers,

We would like to thank you for reviewing our manuscript and for helping us to improve it for possible publication. We agreed with all suggestions and we tried to improve the manuscript accordingly. All changes were highlighted throughout the manuscript and a point-by-point response is given below:

Reviewer 2:

The title of your work is: "Assessment of trabecular bone for dental implant planning using magnetic resonance imaging" but in the paper the informations regarding the planning of dental implants are few. Could you consider to improve this part o modify the title?

We modified the title (lines 1,2) as described below:

“Trabecular bone assessment using magnetic resonance imaging: a pilot study”

Please check again the typos and form, ex. line 117 missing bracket in bib.

Response:

We corrected all references throughout the text.

I suggest to expand the field of this pilot study, introducing some evaluation of TMJ, where the MRI is widely used, you can use the following paper as reference: Sambataro, S.; Cervino, G.; Bocchieri, S.; La Bruna, R.; Cicciù, M. TMJ Dysfunctions Systemic Implications and Postural Assessments: A Review of Recent Literature. J. Funct. Morphol. Kinesiol. 2019, 4, 58.

Response:

We kept the initial focus of the study, since this is a continuity of a previous published study (see line 52). However, we discussed the possibility to expand these results to further fields, as the referenced paper (lines 140-144):

“Although this analysis was focused on dental implant planning, where a three-dimensional bone evaluation is required [11,12,23], assessment of bone microarchitecture is well-known in medical literature. For instance, MRI has been shown to be a technique for monitoring systemic bone diseases, since it allows recognizing changes on trabecular bone morphology [22]. Thus, developing specific protocols for monitoring of bone remodeling processes could benefit different medical fields [30,31].”

Please expand the conclusion, including some clinical implication of this work.

Response:

We improved the discussion (lines 153-155):

“Within the limitation of this study, MRI overestimated trabecular bone parameters but with a statistically significant fixed linear offset. Further studies are required to determine the clinically feasibility of MRI for trabecular bone assessment. “

Reviewer 3 Report

Dear authors, you performed a study about potential use of Magnetic Resonance imaging for the assessment of trabecular bone morphology. Although the sample size is very small, the results are of interest for dental implant specialist. Methods and statistical analyses are well explained. However, discussion could be improved, including more information about the state of the art, strength and limitations of Magnetic Resonance in field of dental implant. Moreover, it could be interesting provide a mention at the end of discussion how pharmacological and nutraceutical treatments could influence the evaluation of the proposed radiological parameters. For this purpose, i suggest to read and cite "Nastri L, Moretti A, Migliaccio S, Paoletta M, Annunziata M, Liguori S, Toro G, Bianco M, Cecoro G, Guida L, Iolascon G. Do Dietary Supplements and Nutraceuticals Have Effects on Dental Implant Osseointegration? A Scoping Review. Nutrients. 2020 Jan 20;12(1):268. doi: 10.3390/nu12010268. PMID: 31968626; PMCID: PMC7019951."

Author Response

Dear reviewers,

We would like to thank you for reviewing our manuscript and for helping us to improve it for possible publication. We agreed with all suggestions and we tried to improve the manuscript accordingly. All changes were highlighted throughout the manuscript and a point-by-point response is given below:

Reviewer 3:

Dear authors, you performed a study about potential use of Magnetic Resonance imaging for the assessment of trabecular bone morphology. Although the sample size is very small, the results are of interest for dental implant specialist. Methods and statistical analyses are well explained. However, discussion could be improved, including more information about the state of the art, strength and limitations of Magnetic Resonance in field of dental implant. Moreover, it could be interesting provide a mention at the end of discussion how pharmacological and nutraceutical treatments could influence the evaluation of the proposed radiological parameters. For this purpose, i suggest to read and cite "Nastri L, Moretti A, Migliaccio S, Paoletta M, Annunziata M, Liguori S, Toro G, Bianco M, Cecoro G, Guida L, Iolascon G. Do Dietary Supplements and Nutraceuticals Have Effects on Dental Implant Osseointegration? A Scoping Review. Nutrients. 2020 Jan 20;12(1):268. doi: 10.3390/nu12010268. PMID: 31968626; PMCID: PMC7019951."

Response:

We thank you for your valuable comments and suggestions. We improved the discussion regarding the clinical application of magnetic resonance in dental implant therapy, such as in further fields involving bone remodeling processes.

Reviewer 4 Report

Title. The terms "Pilot Study" should be included in the title.

Introduction. New CBCT provides very low irradiation, therefore please be careful in stating that CBCT could be harmful to patients.

Introduction. Do the authors think that, for implant treatment planning, bone quality is more important than bone quantity?

Materials and Methods. Was porcine bone used due to its similarity to human bone? Please clarify this aspect.

M&M. It could be of interest to know the mean age of the animal samples used in this study. Moreover, where this samples maxillary or mandibular bone?

Discussion. Please insert reference 20 in the correct Journal form.

Discussion. The discussion part is extremely poor and concise. Please discuss the study results at least with other references already cited into the Introduction part (8,9,11-19) concerning the use of MRI for implant planning.

Author Response

Dear reviewers,

We would like to thank you for reviewing our manuscript and for helping us to improve it for possible publication. We agreed with all suggestions and we tried to improve the manuscript accordingly. All changes were highlighted throughout the manuscript and a point-by-point response is given below:

Reviewer 4:

Title. The terms "Pilot Study" should be included in the title.

Response:

We changed the title accordingly (lines 1,2):

“Trabecular bone assessment using magnetic resonance imaging: a pilot study”

Introduction: New CBCT provides very low irradiation, therefore please be careful in stating that CBCT could be harmful to patients.

Response:

We agree with your observation, and we deleted this statement to avoid missinterpretation.

Do the authors think that, for implant treatment planning, bone quality is more important than bone quantity?

Response:

No, we believe that both analysis can complement and improve the treatment planning. We rewrite this statement accordingly (lines 30-37):

As far as the dental implant planning is concerned, special attention has been given to trabecular bone assessment. Although an adequate bone volume is a requisite for periimplant health, it has been proved that bone quality may also play an important role on dental implant outcomes. According to the Classification of Lekholm and Zarb [1], jaw bone may either present cortical bone with different thickness or trabecular bone with variable microarchitecture. In this sense, recent studies have already shown that cortical thickness influences the primary stability of dental implants [2,3]. However, a successful implant treatment is related not only to bone volume, but also to micromorphology of trabeculae and how they are arranged and connected to each other [4].“

Materials and Methods. Was porcine bone used due to its similarity to human bone? Please clarify this aspect.

Yes, based on previous studies, we used porcine ribs because of its similarity with maxillary human bone. We added this explanation on the discussion (line 128):

Porcine ribs were chosen due to its similarity with human maxillary bone [26].”

M&M. It could be of interest to know the mean age of the animal samples used in this study. Moreover, where this samples maxillary or mandibular bone?

Response:

Unfortunately we can not say how old the animals were when we acquired the samples. We used porcine ribs freshly acquired, and we added this detail on “Material and Methods” (line 52):

“Porcine ribs acquired on a local butchery for a previous study [26] were used on this study.”

Discussion. Please insert reference 20 in the correct Journal form.

Response:

We corrected all references throughout the manuscript.

Discussion. The discussion part is extremely poor and concise. Please discuss the study results at least with other references already cited into the Introduction part (8,9,11-19) concerning the use of MRI for implant planning.

Response:

We tried to improved the discussion by discussing the previous literature, showing the limitations of this study, its clinical relevance and recommendations for future studies (see Discusion, lines 116-151).